# Corneal Edema in Inducible *Slc4a11* Knockout Is Initiated by Mitochondrial Superoxide Induced Src Kinase Activation

**DOI:** 10.3390/cells12111528

**Published:** 2023-06-01

**Authors:** Diego G. Ogando, Edward T. Kim, Shimin Li, Joseph A. Bonanno

**Affiliations:** Vision Science Program, School of Optometry, Indiana University, Bloomington, IN 47405, USA; digogand@indiana.edu (D.G.O.); edtkim@indiana.edu (E.T.K.); shimli@indiana.edu (S.L.)

**Keywords:** corneal endothelial dystrophy, *Slc4a11*, barrier function, lactate transporters, Src kinase, oxidative stress, Visomitin

## Abstract

Purpose: Inducible *Slc4a11* KO leads to corneal edema by disruption of the pump and barrier functions of the corneal endothelium (CE). The loss of Slc4a11 NH_3_-activated mitochondrial uncoupling leads to mitochondrial membrane potential hyperpolarization-induced oxidative stress. The goal of this study was to investigate the link between oxidative stress and the failure of pump and barrier functions and to test different approaches to revert the process. Methods: Mice which were homozygous for Slc4a11 Flox and Estrogen receptor –Cre Recombinase fusion protein alleles at 8 weeks of age were fed Tamoxifen (Tm)-enriched chow (0.4 g/Kg) for 2 weeks, and controls were fed normal chow. During the initial 14 days, Slc4a11 expression, corneal thickness (CT), stromal [lactate], Na^+^-K^+^ ATPase activity, mitochondrial superoxide levels, expression of lactate transporters, and activity of key kinases were assessed. In addition, barrier function was assessed by fluorescein permeability, ZO-1 tight junction integrity, and cortical cytoskeleton F-actin morphology. Results: Tm induced a rapid decay in Slc4a11 expression that was 84% complete at 7 days and 96% complete at 14 days of treatment. Superoxide levels increased significantly by day 7; CT and fluorescein permeability by day 14. Tight junction ZO-1 distribution and the cortical cytoskeleton were disrupted at day 14, concomitant with decreased expression of Cldn1, yet with increased tyrosine phosphorylation. Stromal lactate increased by 60%, Na^+^-K^+^ ATPase activity decreased by 40%, and expression of lactate transporters MCT2 and MCT4 significantly decreased, but MCT1 was unchanged at 14 days. Src kinase was activated, but not Rock, PKCα, JNK, or P38Mapk. Mitochondrial antioxidant Visomitin (SkQ1, mitochondrial targeted antioxidant) and Src kinase inhibitor eCF506 significantly slowed the increase in CT, with concomitant decreased stromal lactate retention, improved barrier function, reduced Src activation and Cldn1 phosphorylation, and rescued MCT2 and MCT4 expression. Conclusions: Slc4a11 KO-induced CE oxidative stress triggered increased Src kinase activity that resulted in perturbation of the pump components and barrier function of the CE.

## 1. Introduction

Homozygous and compound heterozygous mutations in Solute Linked Cotransporter A11 (*SLC4A11*) cause congenital hereditary endothelial dystrophy (CHED) [1], which is marked by corneal edema and eventual endothelial cell death in early childhood. SLC4A11 is an NH_3_-sensitive electrogenic H^+^ transporter localized to the basolateral membrane and the inner mitochondrial membrane of corneal endothelial cells [2,3,4]. Basolateral plasma membrane SLC4A11 is postulated to contribute to the corneal endothelial pump function by an H^+^ buffering mechanism that enhances lactate transport [5,6]. More significantly, inner mitochondrial membrane SLC4A11 is an NH_3_-sensitive mitochondrial uncoupler that is activated by glutamine (Gln) catabolism [7,8]. Gln enters the tricarboxylic acid cycle, yielding a significant increase in reducing potential that drives the electron transport chain and ATP production. Slc4a11 provides a mild uncoupling, preventing mitochondrial membrane potential (MMP) hyperpolarization, which suppresses superoxide production [3]. In *Slc4a11*-deficient mouse corneal endothelial cells, Gln hyperpolarizes mitochondria, increasing superoxide and opening the permeability transition pore [5], which depolarizes MMP and induces apoptosis [3]. Inhibiting superoxide production by reducing ammonia production and treating with a mitochondrial uncoupler or direct mitochondrial antioxidant treatment prevented cell death in the presence of glutamine, indicating that mitochondrial superoxide production was the primary factor in inducing cell death [3,9].

In conventional *Slc4a11* KO mice, corneal edema is observed early (<12 weeks of age) with mild changes in cell morphology, yet there is no significant difference in corneal endothelial cell density compared to wild-type mice [10,11]. However, at older ages, cell morphology becomes more aberrant and cell density significantly decreases in the KO [10,11]. In *Slc4a11* KO mice, the gene is null during embryonic development and edema is present at eye-opening, which precludes uncovering the sequence of early events leading to edema. To circumvent this problem, we generated an inducible Slc4a11 KO mouse model that recapitulates the conventional KO phenotype over time [12]. In addition to mitochondrial superoxide production, there are losses in Na^+^-K^+^ ATPase activity and expression of key lactate transporters (monocarboxylate cotransporter, MCTs) in this KO. Moreover, the changes in cell morphology were concomitant with disruption of the barrier function and alteration of the tight junction and cortical cytoskeleton structures [12].

The goal of the current study was to investigate the earliest events within corneal endothelial cells that ultimately lead to edema. Our working hypothesis is that mitochondrial oxidative stress is the initial trigger that leads to alterations in gene expression, cell morphology, and permeability, resulting in corneal edema and eventual cell death. Following induction of Slc4a11 KO, corneal thickness was continually measured, and stromal [lactate], endothelial mitoROS, lactate transporter (MCTs) and Na^+^-K^+^ ATPase expression were measured at selected time points. Since oxidative stress is known to activate Rho Kinase (Rock) and p38MAP Kinase, which mediate disruption of the barrier function in corneal endothelial cells in vitro [13], and since in other cell types, alteration of the cell junctions is mediated by the c-Src, PKCα, and JNK kinases [14,15,16,17,18], we examined a panel of kinase expression and phosphorylation status.

## 2. Materials and Methods

### 2.1. Drugs

All drugs, i.e., Visomitin, #HY-100474; eCF506, #HY-112096; and Ripasudil, #HY-15685, were obtained from MedChemExpress (Monmouth Junction, NJ, USA).

### 2.2. Mice and Therapies

The generation of the inducible *Slc4a11* KO mouse was described previously [12]. *Slc4a11* Flox/Flox mice were generated by Ozgene (Indianapolis, IN, USA) [10]. *Cre-ERT2* mice (Stock No: 008463) were obtained from Jackson Laboratories (Bar Harbor, ME, USA). In this strain, ubiquitous Cre-ERT2 expression is directed by a strong promoter. *Slc4a11^Flox/Flox^*//*Rosa^CreERT2^^/CreERT2^* mice, at 8 weeks of age, were fed with Tamoxifen (Tm)-enriched chow (0.4 g/kg, Envigo #TD130859, Indianapolis, IN, USA) for two weeks, followed by normal chow. Whole-body *Slc4a11* KO is expected in this model. The genotyping procedures for *Slc4a11* flox and for *CreERT2* alleles were described previously [12]. The non-specific effects of Tamoxifen on corneal thickness were insignificant [12]. Central corneal thickness was measured by optical coherence tomography (iVue100 Optovue, Inc., Fremont, CA, USA) as described in detail previously [12]. Eye drop therapy was performed as follows: using Visomitin (SkQ1) 1.5 µM in PBS and eCF506 1 µM in PBS or PBS, 10 µL drops were applied thrice/day (at 9 a.m., 1 p.m., 5 p.m.) in both eyes to *Slc4a11^Flox/Flox^*//*CreERT2/CreERT2* mice at 8 weeks of age, starting at the same time as Tamoxifen feeding. In addition, central corneal thickness was measured prior to and at 14 days of treatment. Mice were euthanized by inhalation of carbon dioxide followed by cervical dislocation at different time points, and corneas were obtained for analysis. Using both OCT and slit-lamp examination, no corneal ulcers, surface defects, or inflammation were observed in any of the groups of mice. All mice were housed in pathogen-free conditions and used in the experiments in accordance with institutional guidelines and the current regulations of the National Institutes of Health, the United States Department of Agriculture, and the Association for Research in Vision and Ophthalmology (ARVO) Statement for the Use of Animals in Ophthalmic and Vision Research. All experiments involving mice were conducted in accord with our approved IACUC protocol from Indiana University Bloomington, as was the number of mice used.

### 2.3. QPCR

QPCR was performed as described previously [12]. Briefly, corneal endothelium-Descemet’s membrane (CEDM) was dissected from whole corneas using jeweler’s forceps from three mice, which were pooled to obtain one sample. Three Ctrl and three Tm samples were subjected to QPCR. Slc4a11 primers: forward: CTGTGAGGTTCGCTTTGT; reverse: GTGCCAGTCTTCAGGAGC. Beta actin primers: CTAAGGCCAACCGTGAAA; reverse: ACCAGAGGCATACAGGGA.

### 2.4. Stromal Lactate

Corneas were obtained, and the epithelium and endothelium were removed. Individual stromas were placed in pre-weighed Eppendorf tubes, pulverized in liquid nitrogen, and homogenized in 30 µL of PBS using a plastic disposable pestle. The sample was centrifuged at 15,000× *g* for 15 min at 4 °C. The supernatant was recovered. The remaining pellet was dried at 60 °C in a vacuum centrifuge for two hours, then weighed (dry weight). Lactate was measured in the supernatant, n ≥ 3 for Ctrl and Tm, using a fluorescent kit (Abcam #ab65330, Cambridge, UK) according to the manufacturer’s instructions.

### 2.5. Endothelial Permeability Assay

Whole corneas were dissected and placed into small dimples on plates, endothelial side up. A volume of 10 µL of 0.1% sodium fluorescein in bicarbonate-rich Ringer (BR) was added on the endothelial side and incubated at room temperature for 30 min. The corneas were washed thrice for five minutes each with 10 µL BR. After removing all liquid, corneas were positioned in wells of a 96-well plate, and fluorescence (excitation: 485 nm, emission: 520 nm) was measured in a microplate reader. As a positive control (n ≥ 3 for Ctrl and Tm), corneas were incubated for one hour with 10 µL EGTA 2 mM in Ca-free PBS before incubation in fluorescein.

### 2.6. Na^+^-K^+^ ATPase Activity

Two corneal endothelial-Descemet membrane (CEDM) peelings from the same mouse were pooled and homogenized in 30 µL assay buffer, provided in the ATPase Assay kit (Abcam #ab234055), using a plastic disposable pestle. After sonication, the sample was centrifugated at 10,000× *g* for 10 min at 4 °C. The supernatant was recovered, and the phosphates in the sample were depleted by incubation in 40 µL of PiBind resin (Innova Biosciences #501-0015, Montluçon, France) for 15 min at room temperature in a rotary device. After centrifugation at 1000× *g* for two minutes, the sample was recovered, and ATPase activity was measured in 5 µL of sample in the presence or absence of 1 mM ouabain. Na^+^-K^+^ ATPase activity (n ≥ 3 for Ctrl and Tm) was obtained by subtracting the activity in presence of ouabain from the total activity. Protein was measured using the BCA method.

### 2.7. Mitochondrial Superoxide

Following euthanasia, corneas were dissected, maintained in Hanks Balanced Salt Solution at 37 °C, and stained with MitoSOX (Thermo Fisher Scientific #M36008, Waltham, MA, USA) 1 µM in HBSS for 30 min at 37 °C, then washed 3 times for 5 min in HBSS. To facilitate the identification of the endothelial cells, nuclei were stained with Hoechst 1 µg/mL for 10 min at room temperature and then washed 3 times for 5 min in HBSS. Corneas were flattened by performing 4 relaxing cuts, then positioned with the endothelial surface facing down in a glass-bottom Petri Dish (MetTek #P35G-1.5-20-C, Lucknow, ON, Canada) in 100 µL of HBSS. An 18 mm coverslip was placed over the cornea, with a plastic weight to flatten the tissue. Images were taken with a Zeiss Axio Observer Z1 inverted microscope (Zeiss, Pleasanton, CA, USA) microscope using a 40× objective. Five images were taken for each cornea. Using Image J, the mean fluorescence intensity of each individual cell was obtained and then averaged per image and cornea (n = 4 for Ctrl and Tm).

### 2.8. Immunofluorescence

Immunofluorescence was performed as previously described [12]. The primary antibody was mouse anti-ZO-1 (1:100) (Thermo Fisher Scientific #33-9100). Phalloidin-A488 (Thermo Fisher Scientific #A12379) 1× solution was used to stain F-actin.

### 2.9. Immunoprecipitation

Immunoprecipitation was performed following instructions from the Pierce Classic Immunoprecipitation kit (Thermo Fisher Scientific #26146). Briefly, two CEDM from one mouse were pooled and lysed in 30 µL of IP lysis buffer supplemented with protease plus phosphatase inhibitors. Five µL of lysate as saved to use as input. In a spin column provided by the kit, 25 µL of lysate was combined with 475 µL of IP lysis buffer and pre-cleared with 20 µL of control agarose slurry for one hour at 4 °C in a rotary device. The immune complex was prepared by combining the 500 µL of pre-cleared diluted lysate with 1 µg of rabbit P-tyrosine antibody (Cell Signaling Technology #8954S, Danvers, MA, USA), 1 µg of control rabbit IgG (Cell Signaling Technology #3900S), or 1 ug of control mouse IgG (Cell Signaling Technology #5415S) and incubating for 18 h at 4 °C in a rotary device. The immune complex was captured by incubating the antibody/lysate sample with 20 µL of protein A/G agarose slurry for one hour at 4 °C in a rotary device. The agarose containing the immune complex was washed four times with 200 µL of IP lysis buffer and once with 1× conditioning buffer supplied by the kit. The complex was eluted by adding 10 µL of 2× Mastermix from the Protein Simple Jess kit and then boiling for 10 min. After centrifugation from 1000× *g* for 1 min, the eluted protein was recovered and used to run the Western immunoassay for Cldn1 and ZO-1 (n = 3 for Ctrl and Tm), as described below.

### 2.10. Western Immunoassay

Two CEDM peelings from one mouse were pooled and lysed in 25 µL RIPA lysis buffer containing protease plus phosphatase inhibitors. Protein was measured using the BCA method. Equal amounts of protein (1.5 µg) were loaded into wells of the 12 to 230 kDa separation module of a Protein Simple Jess system (Protein Simple, San Jose, CA, USA). Antibodies were added at the following dilutions: MCT1 (Abcam #ab90582) 1:10; MCT2 (Santa Cruz Biotechnologies #sc-166925, Santa Cruz, CA, USA) 1:10; MCT4 (Santa Cruz #sc-376465) 1:10; Atp1a1 (Abcam #ab760020) 1:10; Atp1b2 (Abcam #ab185210) 1:10; ZO-1 (Thermo Fisher Scientific #33-9100) 1:10; Cldn1 (Thermo Fisher Scientific #37-4900) 1:10; β-catenin (Cell Signaling Technology #9582) 1:10, P(S19)-MLC (Cell Signaling Technology #3671) 1:10; P-PKCα (Cell Signaling Technology #9375) 1:10; P-JNK (Cell Signaling Technology #9255S) 1:10; P-P38Mapk (Cell Signaling Technology #9215S) 1:10; Src (Cell Signaling Technology #2109) 1:50; and P-Src (Cell Signaling Technology #6943) 1:10. Secondary antibodies and the substrate were provided by the Jess kit. For normalization, the total protein on each lane was quantified with the Total Protein Module (Protein Simple #DM-TP01). Jess data were obtained (n ≥ 3 for Ctrl and Tm) as virtual blots in which molecular weight and signal intensity were presented. Results, in the form of a traditional electropherogram, were also obtained with this approach.

### 2.11. Statistical Analysis

All corneal thickness measurements had at least 5 replicates. All other assays had at least 3 replicates. Results are presented as mean ± SEM. Statistical analysis was performed with GraphPad 9.4 (GraphPad Software, Inc., Boston, MA, USA). For comparisons between two groups, Student’s *t*-test was utilized. For three or more groups, one-way or two-way ANOVA was performed followed by Tukey’s multiple comparison test. Significance was defined as *p* < 0.05.

## 3. Results

To study the early events leading to corneal edema, Slc4a11 Flox/Flox//CreERT2/CreERT2 mice at 8 weeks of age were fed with tamoxifen (Tm) chow to induce *Slc4a11* knock-out, and different parameters were studied at several time points. Tm induced 84% and 96% Slc4a11 messenger RNA knockdown in the endothelium at 7 and 14 days of Tm treatment, respectively (Figure 1A). The corneal thickness was slightly higher at 7 days, but increased significantly by 14 days post-Tm induction (Figure 1B). Since the underlying mechanism of the endothelial pump was lactate-linked water flux [6], we measured corneal [lactate]. Lactate accumulation in the stroma indicates inhibition of the corneal endothelial pump function. We found that lactate increased significantly at 7 days and remained higher than that in the control group at 14 and 28 days of Tm induction (Figure 1C). Consistent with pump failure, there was a 40% reduction in Na^+^-K^+^ ATPase activity at 14 days post-Tm treatment (Figure 1D).

At 14 days, we examined the expression of Na^+^-K^+^ ATPase components and the lactate transporter MCTs (monocarboxylate cotransporters). Whereas there was no significant change in the expression of subunits Atp1a1 and Atp1b2 (Figure 2A,B), and no changes in MCT1 (Figure 2C), significant decreases in MCT2 and 4 were observed (Figure 2D,E).

While the pump function consists of vectorial (stroma to anterior chamber) transport of lactate and water, maintenance of transendothelial osmotic gradients derived from the ion transport activity requires an intact osmotic barrier that is conferred by tight and adherens junctions. Therefore, we examined the endothelial permeability and the protein expression associated with barrier function. The integrity of the tight junctions (studied by ZO-1 staining) and the cortical cytoskeleton (studied by F-actin) were altered at 14 days, but this was not noticeable at 7 days (Figure 3A). Similarly, endothelial permeability increased at 14 days, but not at 7 days (Figure 3B). To explore potential changes in tight junction assembly that may explain the increase in permeability, we examined the expression of several junction proteins. As is consistent with the increased permeability at 14 days, Cldn1 levels were decreased, whereas ZO-1 levels were increased in Tm-treated mice, while β-catenin levels were not changed (Figure 3C).

We hypothesized that a trigger for these changes in protein expression and subsequent corneal edema was the oxidative stress caused by Slc4a11 deletion [3,19]. We observed that mitochondrial superoxide levels increased in the Tm group after 7 days when significant Slc4a11 knockdown occurred (Figure 4A,B). Mitochondrial superoxide levels remained elevated at 14 days (Figure 4A,B). Thus, mitoROS production precedes the development of corneal edema.

Several kinases have been associated with ROS-induced disruption of the barrier function: the Rock, MLCK, PKCα, JNK, c-SRC, and p38 MAP kinases [14,17,18,20,21,22]. None of the ROCK, MLCK (tested by phosphorylation of MLC), PKCα, JNK, or p38 MAP kinases were activated by Tm at 14 days (Figure 5A–D). Only c-Src was activated at 14 days (Figure 5E). While ZO-1 tyrosine phosphorylation decreased in the Tm samples, (Figure 5F), Cldn1 tyrosine phosphorylation increased (Figure 5G), although the total expression decreased (Figure 3C).

ROCK inhibitor topical therapy has been used to treat Fuchs’ endothelial corneal dystrophy (FECD) [15]. Rho kinase activation can directly or indirectly result in MLC phosphorylation, leading to actomyosin contraction and disruption of the barrier function [23]. We did not find increased MLC phosphorylation in our model. However, ROCK activation can be detrimental to the barrier and the pump function by mechanisms independent of increasing MLC phosphorylation [24]. For example, ex vivo treatment of the corneal endothelium with Rho kinase inhibitor Ripasudil increased the expression of proteins of the pump and barrier function [24]. For these reasons, we tested topical treatment with Ripasudil. Ripasudil eye drops thrice per day had no effect on the corneal thickness, fluorescein permeability, tight junctions, or cortical cytoskeleton (Figure 6A–C).

With evidence for the induction of mitochondrial oxidative stress and c-Src kinase activation, we used Visomitin to reduce oxidative stress and eCF506 (Src kinase inhibitor) to inhibit c-Src kinase activity directly in the mouse model via topical delivery. In both cases, eye drop therapy significantly reduced the increase in corneal thickness induced by *Slc4a11* knockout, indicating that mitochondrial ROS and Src kinase mediated the edema (Figure 7A). In addition, both drugs significantly inhibited the increase in permeability (Figure 7B) and lactate accumulation (Figure 7C) induced by *Slc4a11* knockout. Moreover, topical Visomitin and eCF506 rescued tight junction (ZO-1) and opposed cortical cytoskeleton (F-actin) disorganization induced by *Slc4a11* knockout (Figure 7D). Furthermore, Visomitin and eCF506 significantly decreased Src phosphorylation in the KO relative to control PBS drops (Figure 8A), indicating that mitochondrial oxidative stress triggers Src kinase activation. Both drugs reversed the decrease in Cldn1 levels and decreased Cldn1 tyrosine phosphorylation (Figure 8B,C). The drugs also recovered Slc4a11 KO-induced decline in MCT2 (Figure 8D) and MCT4 (Figure 8E), and had no effect on MCT1 levels (Figure 8F).

## 4. Discussion

In the current study, we found that mitochondrial oxidative stress precedes the development of edema in the inducible *Slc4a11* KO. Pump function is altered by decreased levels of MCT2 and MCT4 and reduced Na^+^-K^+^ ATPase activity, leading to lactate accumulation in the stroma. Furthermore, barrier function is altered, as indicated by the elevated fluorescein permeability, the breakdown of the tight junction and cortical cytoskeleton, the reduction in Claudin 1 levels, and the increased Cldn1 tyrosine phosphorylation via activation of the c-Src kinase. MitoROS (i.e., superoxide) and Src Kinase appeared to be the main players in the onset of edema, as they were inhibited by Visomitin and eCF506.

Oxidative stress is the main contributor to the pathophysiological process of FECD and CHED [25,26,27]. In the *Slc4a11* KO, the lack of NH_3_-activated mitochondrial uncoupling hyperpolarizes the mitochondrial membrane potential, leading to excessive superoxide production [3]. In the inducible KO, Visomitin eye drop therapy successfully reverted the edema, indicating that MitoROS is the primary cause of the phenotype. This is consistent with previous data showing a reduction in corneal edema in the conventional KO using MitoQ (a mitochondrial targeted antioxidant) i.p. [9] along with Dimethyl αketoglutarate (bypass glutaminolysis, which reduces MitoROS) topical therapy [3]. Visomitin has been shown to improve dry eye symptoms [28,29], and may be a good candidate for topical treatment in CHED or FECD.

Src is implicated in the tyrosine phosphorylation of ZO-1, Cldns, Ocldn, E-cadherin, N-cadherin, β-catenin, and other proteins of tight and adherens junctions [14,15,16]. Src kinase tyrosine phosphorylation can produce degradation or mislocalization of the junction components [14,15]. We found that Src kinase was activated in the KO concomitant with decreased Cldn1 expression, but increased Cldn1 phosphorylation. The increased fluorescein permeability, the disorganization of the tight junctions and cortical cytoskeleton, the tyrosine phosphorylation, and the decreased levels of Cldn1 were all reverted by Src kinase inhibitor topical therapy. Src kinase is known to be activated upon oxidative stress and to impair the barrier function by tyrosine phosphorylation of tight and adherens junction proteins, resulting in their degradation or mislocalization in colon and bile duct epithelial cells [14,15,16]. Hydrogen peroxide directly activates Src by Tyrosine sulfenylation, which leads to Tyrosine 416 phosphorylation and activation [30], as is consistent with our finding in the corneal endothelium that Src kinase activation disrupts tight junctions.

ZO-1 localization was altered, as observed by immunofluorescence. However, ZO-1 levels increased, as was tested by Western blot. This increase may be a compensatory response to tight junction disruption, as observed for tight junction proteins in intestinal epithelium [31]. ZO-1 tyrosine phosphorylation decreased in *Slc4a11* KO, indicating that the Src kinase is unlikely to drive changes in ZO-1 localization.

The mechanism by which *Slc4a11* KO-induced mitoROS leads to decreased MCT expression is unknown. We did not expect protection of the pump function (MCT2 and MTC4 expression and lactate levels) by Src inhibition. Src can increase TCA cycle and OXPHOS activity, resulting in increased mitochondrial ROS production [32]. Therefore, Src inhibition could have decreased ROS production in our model. While the transcription factor Nrf2 is activated by ROS and stimulates the transcription of anti-oxidative genes, Nrf2 is disabled in Slc4a11 KO [19], possibly due to mitochondrial dysfunction, which may have a secondary effect on MCTs expression.

As is consistent with the lack of activation of ROCK, topical Ripasudil treatment had no protective effect on corneal edema in the Slc4a11 KO mice. Contrary to our observation, Rock and p38 MAP kinase are associated with disruption of the barrier function upon oxidative stress in vitro [13]. Ripasudil has been found to protect the endothelium and promote wound healing after treatment of FECD patients with Descemet stripping only, as well as in patients after cataract surgery [33,34]. ROCK inhibitor Y-27632 increased the efficiency of cell-based FECD therapy [35]. In our study, we found no activation of the ROCK, PKCα, JNK, or p38 MAP kinases in the Slc4a11 KO CHED model.

## 5. Conclusions

Figure 9 summarizes the sequence of events leading up to formation of corneal edema. Slc4a11 deficiency increases the protons’ motive force, producing excess MitoROS. ROS directly activates the Src kinase, which phosphorylates tight and adherens junction proteins, leading to their degradation or mislocalization. Cortical cytoskeleton alterations are induced as a secondary response, and the end result is a perturbation of the barrier function. The mitochondrial antioxidant Visomitin or the Src kinase inhibitor eCF506 can rescue this process. The specific mechanism of impairment of MCT cotransporter expression and Na^+^-K^+^ ATPase activity, a feature of this model [12,36], via oxidative stress is unknown, and will require further study.

## Figures and Tables

**Figure 1 cells-12-01528-f001:**
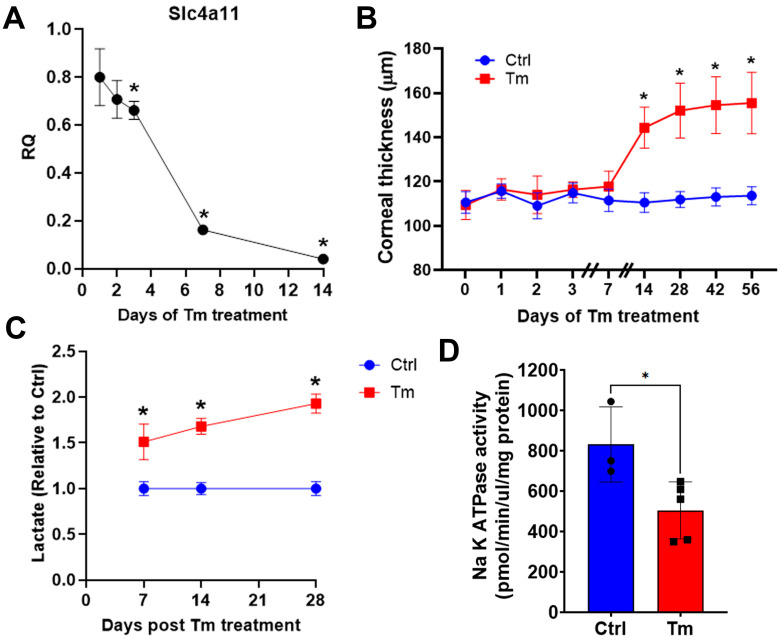
Changes in the pump function in the inducible *Slc4a11* KO. (**A**). Time course of relative Slc4a11 expression by QPCR in CEDM. Relative quantity ± SEM, n = 3, *: *p* < 0.05. (**B**). Corneal thickness time course after Tm treatment; mean ± SEM, n = 5, *: *p* < 0.0001. (**C**). Stomal lactate content; relative values versus control; mean ± SEM, n = 3, *: *p* < 0.05. (**D**). Na^+^-K^+^ ATPase activity at 14 days of Tm treatment; mean ± SEM, n = 3 (Ctrl) and n = 5 (Tm), *: *p* < 0.05.

**Figure 2 cells-12-01528-f002:**
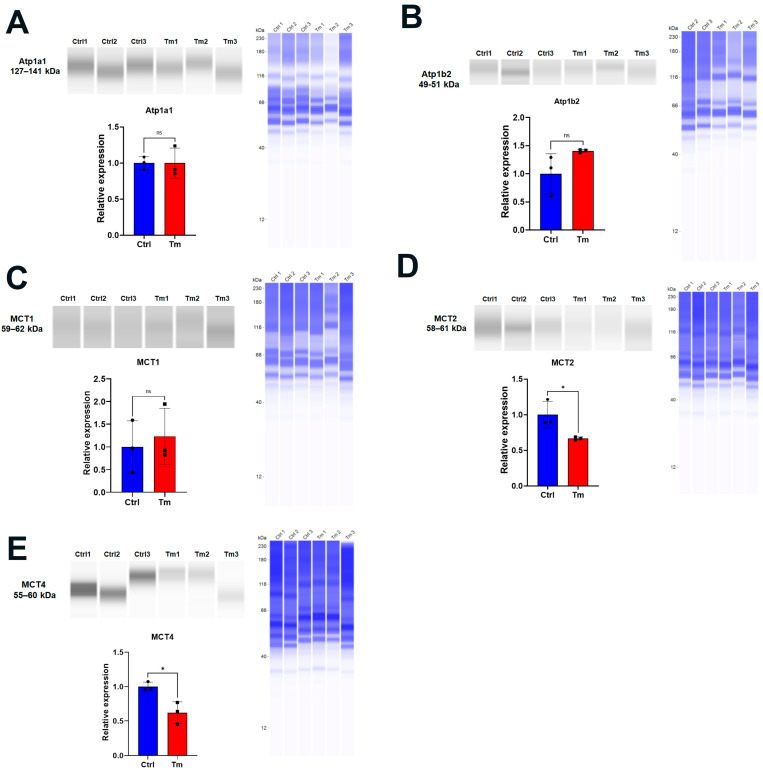
Lactate transporter expression was decreased in the inducible KO at 14 days of Tm treatment. Jess immunoassays: (**A**). Atp1a1. (**B**). ATP1b2. (**C**). MCT1. (**D**). MCT2. (**E**). MCT4. Blue blot shows total protein. Relative values versus control; mean ± SEM, n = 3. *: *p* < 0.05. ns = not significant.

**Figure 3 cells-12-01528-f003:**
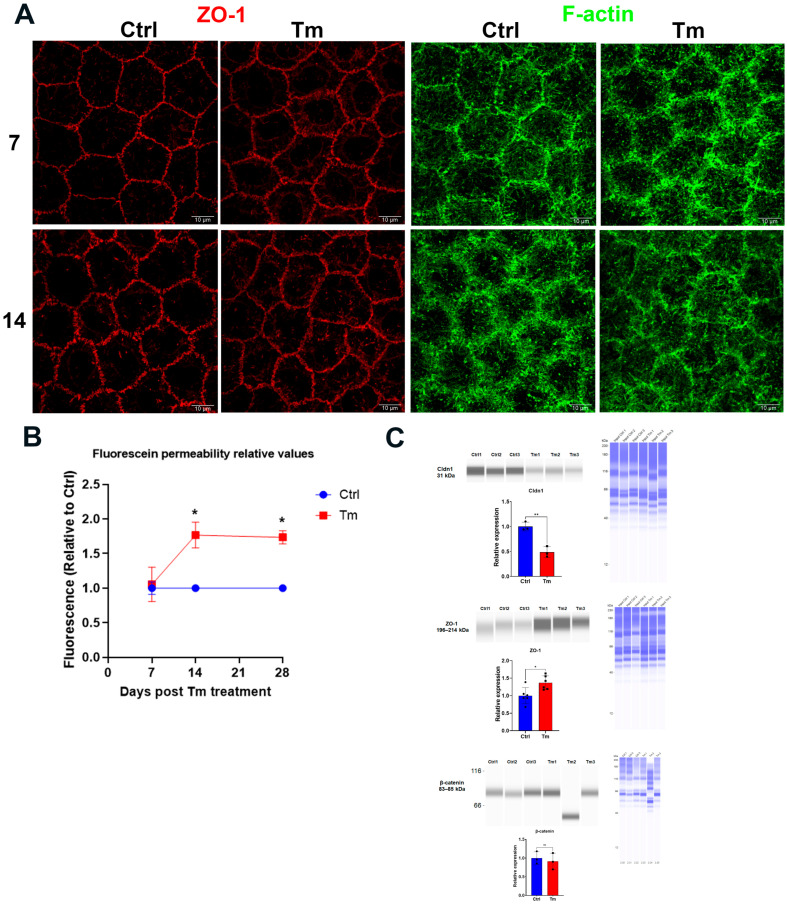
The tight junction and cortical cytoskeleton structure, as well as endothelial permeability, were altered at 14 days of Tm treatment. (**A**). Representative images of ZO-1 and F-actin at 7 and 14 days of Tm treatment. (**B**). Endothelial fluorescein permeability. Relative values versus control; mean ± SEM, n = 3, *: *p* < 0.01. (**C**). Jess immunoassay of ZO-1, Cldn1, and β-catenin. Relative values versus control; mean ± SEM, n = 6 (ZO-1) and n = 3 (Cldn1 and β-catenin). *: *p* < 0.05. **: *p* < 0.01. ns = not significant.

**Figure 4 cells-12-01528-f004:**
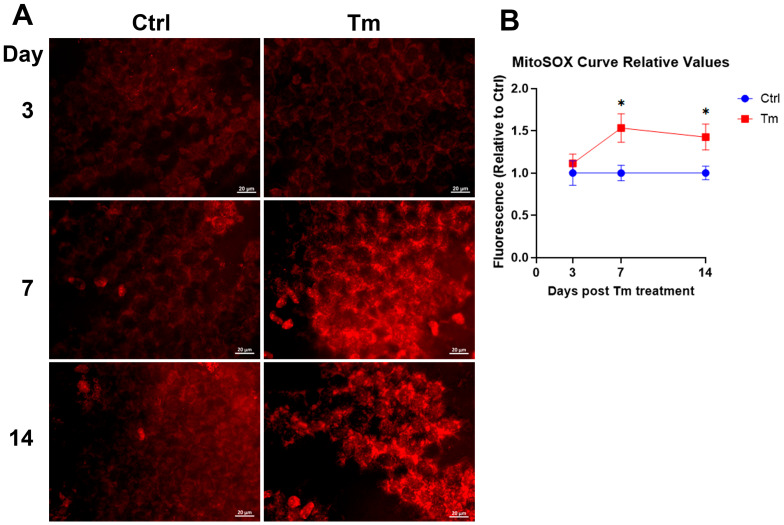
MitoROS coincides with *Slc4a11* knockdown and precedes corneal edema. (**A**). Representative MitoSOX staining at 3, 7, and 14 days of Tm treatment. (**B**). Quantification of MitoSOX staining. Relative values versus control; mean ± SEM, n = 4, *: *p* < 0.05.

**Figure 5 cells-12-01528-f005:**
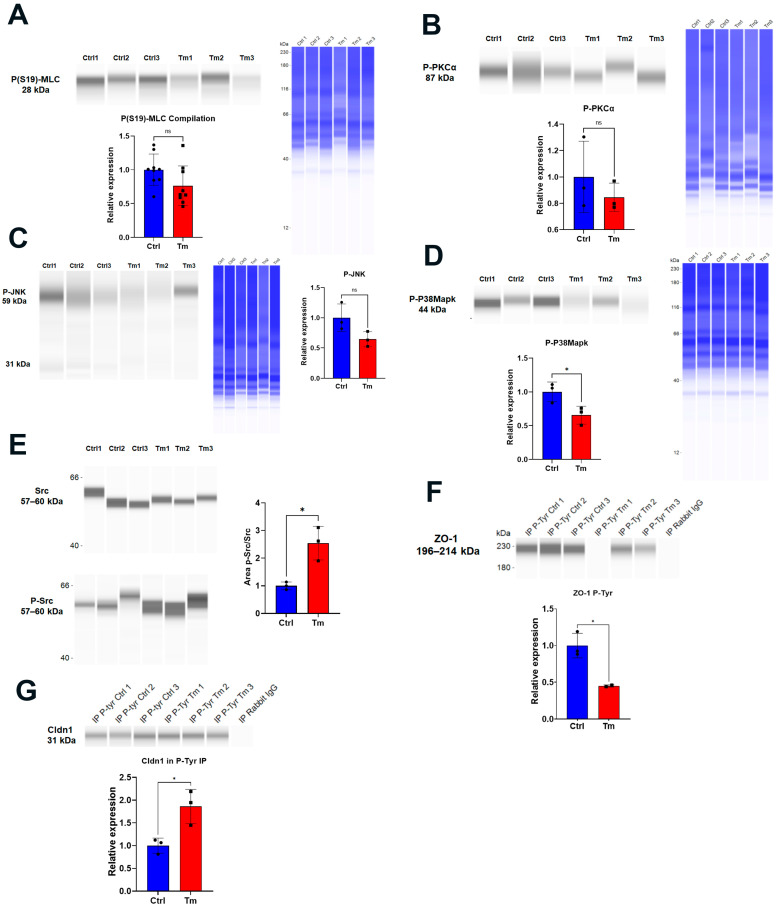
Src kinase was activated and Cldn1 tyrosine phosphorylation increased at 14 days post-Tm treatment. Jess immunofluorescence and quantification of: (**A**) P(S19)-MLC, n = 9; (**B**) P-PKCα, n = 3; (**C**) P-JNK, n = 3; (**D**) P-P38Mapk, n = 3, *: *p* < 0.05.; and (**E**) Src and P-Src, n = 3, *: *p* < 0.05. (**F**) ZO-1 blot from P-tyrosine IP samples: for relative expression calculation, intensity values were normalized to input values found in the middle panel of Figure 2C and then to control values, n = 3, *: *p* < 0.05. (**G**) Cldn1 blot from P-tyrosine IP samples: for relative expression calculation, intensity values were normalized to input values found in the upper panel of Figure 2C and then to control values, n = 3, *: *p* < 0.05. ns = not significant.

**Figure 6 cells-12-01528-f006:**
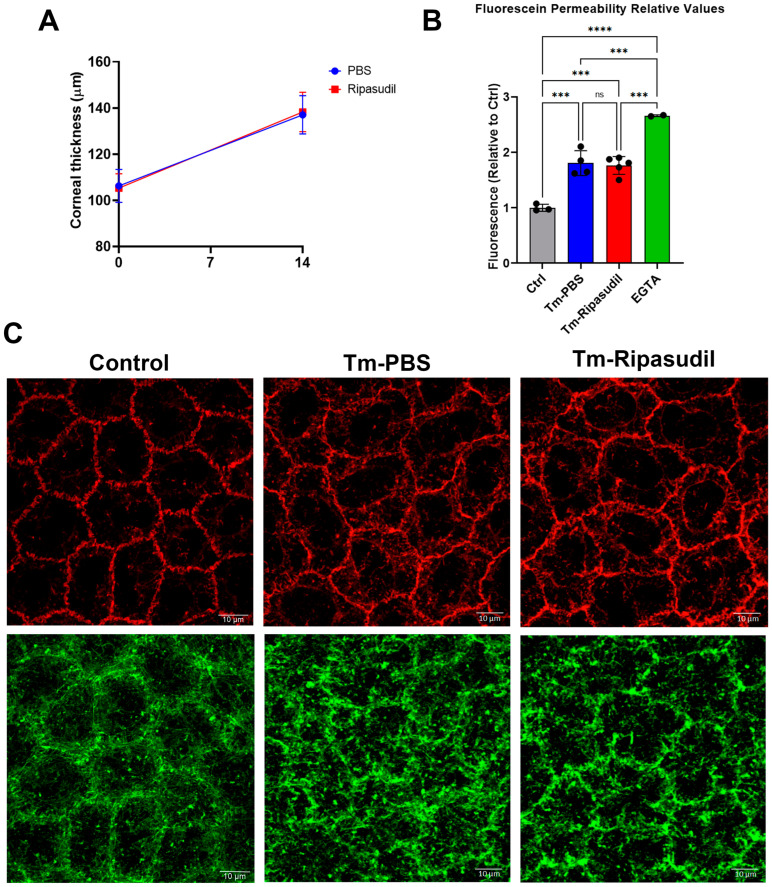
Ripasudil eye drop therapy had no effect on *Slc4a11* KO-induced corneal edema. All measurements were performed at 14 days post-Tm treatment. (**A**). Corneal thickness before and after Ripasudil or PBS eye drop therapy, n = 5 for Ripasudil and n = 4 for control. (**B**). Relative endothelial fluorescein permeability upon eye drop therapy, measured after 14 days of Tm treatment, n = 4 for Tm-PBS, n = 5 for Tm-Ripasudil, n = 3 for control (not treated with Tm), and n = 2 for EGTA. ***: *p* < 0.001. ****: *p* < 0.0001. (**C**). Representative images of ZO-1 and F-actin after 14 days of Ripasudil or PBS eye drop therapy and Tm treatment. ns = not significant.

**Figure 7 cells-12-01528-f007:**
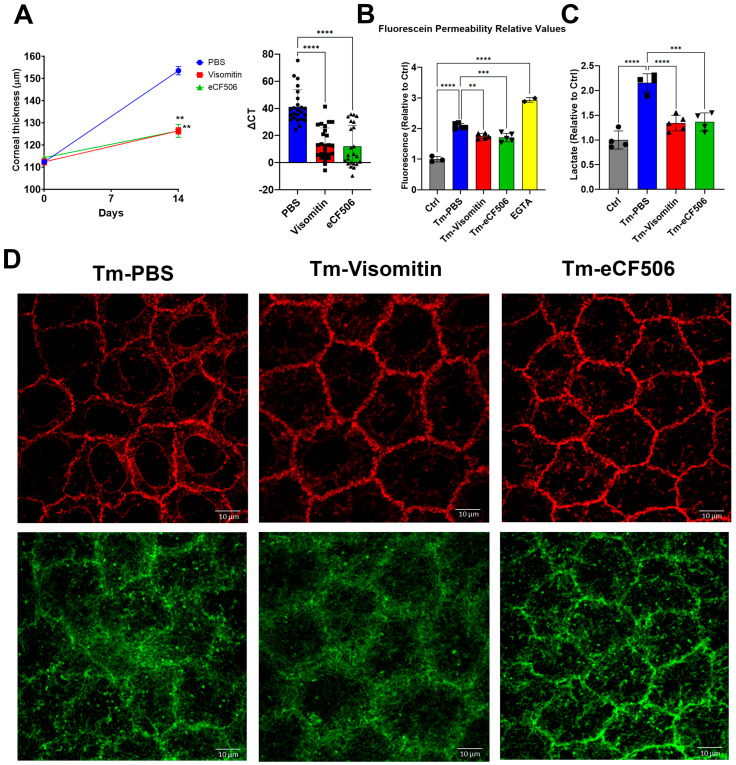
Visomitin and eCF506 eye drop therapies inhibited *Slc4a11* KO-induced corneal edema. (**A**). Corneal thickness and ΔCT after Visomitin, eCF506, or PBS eye drop therapy: n = 12 (24 eyes) for PBS, n = 13 (26 eyes) for Visomitin, and n = 12 (24 eyes) for eCF506. **: *p* < 0.001,****: *p* < 0.0001. (**B**). Relative endothelial fluorescein permeability upon eye drop therapy measured at 14 days of Tm treatment: n = 5 for Tm-PBS, Tm-Visomitin, and Tm-eCF506; n = 3 for control (not treated with Tm); and n = 2 for EGTA. **: *p* < 0.01, ***: *p* < 0.001, ****: *p* < 0.0001 (**C**). Stomal lactate content after 14 days of Tm; relative values versus control; mean ± SEM: n = 4 for control, Tm-PBS, and Tm-eCF506; and n = 5 for Tm-Visomitin. ***: *p* < 0.001, ****: *p* < 0.0001. (**D**). Representative images of ZO-1 and F-actin after 14 days of Visomitin, eCF506, or PBS eye drop therapy and Tm treatment.

**Figure 8 cells-12-01528-f008:**
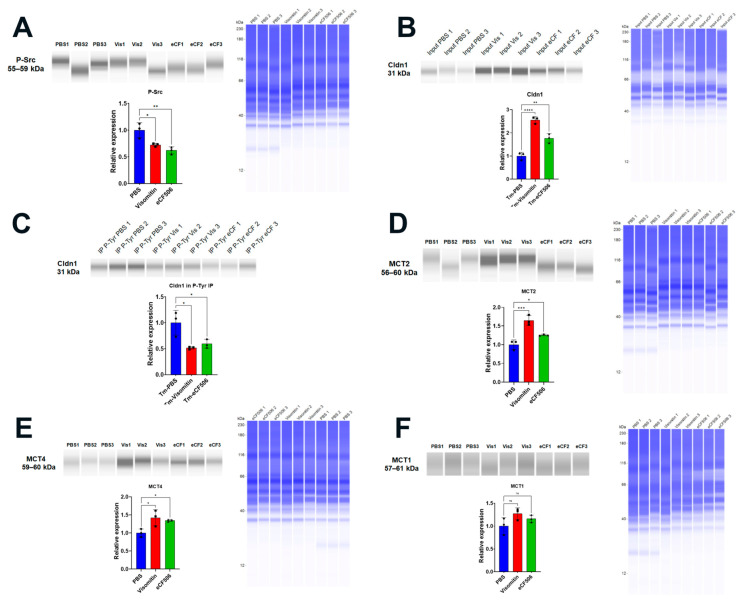
Visomitin and eCF506 eye drop therapy inhibited Src kinase and rescued expression of Cldn1, MCT2, and MCT4 after 14 days of Tm. Jess immunoassays and quantifications are shown. (**A**) P-Src, n = 3, *: *p* < 0.05, **: *p* < 0.01. (**B**) Cldn1, n = 3, **: *p* < 0.01, ****: *p* < 0.0001. (**C**) Cldn1 blot from P-tyrosine IP samples. For relative expression calculation, intensity values were normalized to input values found in B and then to control values, n = 3, *: *p* < 0.05. (**D**) MCT2, n = 3, *: *p* < 0.05, ***: *p* < 0.001. (**E**). MCT4, n = 3, *: *p* < 0.05. (**F**). MCT1, n = 3. ns = not significant.

**Figure 9 cells-12-01528-f009:**
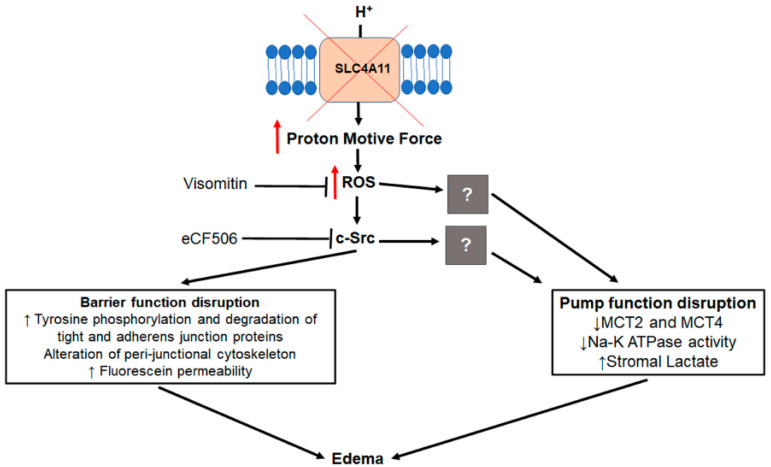
Lack of Slc4a11 leads to increased proton motive force, resulting in increased mitoROS. ROS directly activate Src kinase, phosphorylating tyrosine residues of tight and adherens junction proteins and leading to their degradation and/or mis-localization. Cortical cytoskeleton alterations are induced as a secondary response. The result is perturbation of the barrier function. ROS and Src kinase activation also lead to downregulation of MCT2 and MCT4, as well as decreased Na^+^-K^+^-ATPase activity with subsequent stromal lactate accumulation, resulting in failure of the corneal endothelial pump. The end result is corneal edema.

## Data Availability

Data is contained within the article.

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
