# Peer review of "Corneal Edema in Inducible Slc4a11 Knockout Is Initiated by Mitochondrial Superoxide Induced Src Kinase Activation"

_cells, 2023, doi:10.3390/cells12111528_

Round 1

Reviewer 1 Report

Ogando and co-authors have presented a study where Slc4a11 knockout induces corneal endothelium oxidative stress, which in turn triggers an increase in Src kinase activity, leading to perturbations in the pump components and barrier function of the corneal endothelium. The authors have a solid hypothesis and have conducted numerous experiments to substantiate their hypothesis. However, the current manuscript may be challenging for readers to follow due to the unconventional placement of results in the introduction and the hypothesis and discussion within the results section. The manuscript could be enhanced by improving its structure. Please find my point-by-point comments below:

1. Ensure to provide the full terminology before introducing its abbreviation. Throughout the manuscript, there were multiple instances where abbreviations were used without their corresponding full forms being stated initially.

2. Please ensure that gene names are italicized.

3. The final paragraph of the introduction (lines 68-77) should be relocated to the results or discussion section, with appropriate rephrasing. Typically, readers anticipate the authors' hypothesis or the purpose of the study towards the end of the introduction rather than presenting summarized results.

4. I would suggest expanding the background information in the introduction section rather than including the content currently in the last paragraph. The authors have only elaborated on SLC4A11 and the implications of knocking out the Slc4a11 gene. Additional detail is needed on other concepts. Currently, this information is erroneously placed in the results section.

5. The Materials and Methods section also needs improvement:

5-1) The authors haven't mentioned the IACUC approval in the manuscript.

5-2) The total number of mice used in this study, as well as the number of mice used for each experiment, is unclear. In the figures, 'n' is mentioned, but this need to be clarified in the methods section. The authors should clearly specify the total number of animals used in this study and how many were used for each experiment, in accordance with IACUC-approved protocols.

5-3) In line 80, details regarding the source of each material are insufficient. There's no information about the respective companies, not even clear information about MedChemExpress. Additional details about Ozgene and Jackson Laboratories are also needed, in accordance with the rules of scientific papers. If the authors prefer not to add these details within the main text, they should consider providing its list as supplementary material.

5-4) In line 91, the unit of the Optical Coherence Tomography used in this study should be specified. Likewise, in line 147, the model or specifications of the "Zeiss Observer Z1 microscope" should be detailed.

5-5) In line 92, "Visomitin (SkQ1) 1.5 µm in PBM" is unclear. "µm" doesn't appear to refer to particle size in this context. Please check the unit and clarify its meaning.

6. In the results section, the authors included their hypothesis, purpose of experiments, results, and discussion concurrently. I strongly recommend restructuring the manuscript according to traditional scientific paper formatting. This would significantly improve readability. I suggest moving the background information and hypothesis currently in the results section to the introduction, and relocating some aspects of the results to the discussion section.

7. The authors should clarify whether an ophthalmic examination was performed to rule out other ocular diseases that can cause corneal edema, such as corneal ulcer and anterior uveitis. It is important to mention this aspect in the manuscript.

8. The authors suggest an increase in ZO-1 levels may be a compensatory response to tight junction disruption (line 348). To clarify and support this statement, it would be beneficial to cite references that align with this theory. Currently, without such references, the interpretation of these contrary results remains unclear to the reader.

9. In line 355, the authors mention an unknown kinase or transcription factor. For the benefit of the readers and to add depth to your research, please specify or provide more context regarding the unknown kinase or transcription factor. This could include its role in the mechanism you're studying, previous research relating to it, or why its identity remains unknown.

10. Their discussion section appears quite brief and somewhat speculative without substantial evidence to back up their points. To enhance the clarity and depth of the paper, I suggest they move portions of the discussion found in the results section to the actual discussion section. By doing so, they can elaborate on their results more comprehensively, provide more context and interpretation, and tie their findings more explicitly to existing literature. This would also help to streamline the results section, making it more straightforward for readers to follow.

Author Response

Comments and Suggestions for Authors

#1

Ogando and co-authors have presented a study where Slc4a11 knockout induces corneal endothelium oxidative stress, which in turn triggers an increase in Src kinase activity, leading to perturbations in the pump components and barrier function of the corneal endothelium. The authors have a solid hypothesis and have conducted numerous experiments to substantiate their hypothesis. However, the current manuscript may be challenging for readers to follow due to the unconventional placement of results in the introduction and the hypothesis and discussion within the results section. The manuscript could be enhanced by improving its structure. Please find my point-by-point comments below:

  1. Ensure to provide the full terminology before introducing its abbreviation. Throughout the manuscript, there were multiple instances where abbreviations were used without their corresponding full forms being stated initially.

Thank you for pointing this out. We now added the description of each abbreviation.

  1. Please ensure that gene names are italicized.

This has been corrected.

  1. The final paragraph of the introduction (lines 68-77) should be relocated to the results or discussion section, with appropriate rephrasing. Typically, readers anticipate the authors' hypothesis or the purpose of the study towards the end of the introduction rather than presenting summarized results.

We removed results and conclusions from the Introduction, as suggested.

  1. I would suggest expanding the background information in the introduction section rather than including the content currently in the last paragraph. The authors have only elaborated on SLC4A11 and the implications of knocking out the Slc4a11gene. Additional detail is needed on other concepts. Currently, this information is erroneously placed in the results section.

We have expanded the Introduction to elaborate on topics related to the current study, as suggested.

  1. The Materials and Methods section also needs improvement:

5-1) The authors haven't mentioned the IACUC approval in the manuscript.

IACUC approval is now included.

5-2) The total number of mice used in this study, as well as the number of mice used for each experiment, is unclear. In the figures, 'n' is mentioned, but this need to be clarified in the methods section. The authors should clearly specify the total number of animals used in this study and how many were used for each experiment, in accordance with IACUC-approved protocols.

Now number of animals in each experiment is added to Methods.

5-3) In line 80, details regarding the source of each material are insufficient. There's no information about the respective companies, not even clear information about MedChemExpress. Additional details about Ozgene and Jackson Laboratories are also needed, in accordance with the rules of scientific papers. If the authors prefer not to add these details within the main text, they should consider providing its list as supplementary material.

Now details of the companies are added.

5-4) In line 91, the unit of the Optical Coherence Tomography used in this study should be specified. Likewise, in line 147, the model or specifications of the "Zeiss Observer Z1 microscope" should be detailed.

Model number for OCT and Zeiss microscope are now added.

5-5) In line 92, "Visomitin (SkQ1) 1.5 µm in PBM" is unclear. "µm" doesn't appear to refer to particle size in this context. Please check the unit and clarify its meaning.

“µm” was changed to “µM”.

  1. In the results section, the authors included their hypothesis, purpose of experiments, results, and discussion concurrently. I strongly recommend restructuring the manuscript according to traditional scientific paper formatting. This would significantly improve readability. I suggest moving the background information and hypothesis currently in the results section to the introduction, and relocating some aspects of the results to the discussion section.

As mentioned above the Introduction has been edited. Interpretation of results was moved to the Discussion unless it was necessary to formulate the questions for the next experiment.

  1. The authors should clarify whether an ophthalmic examination was performed to rule out other ocular diseases that can cause corneal edema, such as corneal ulcer and anterior uveitis. It is important to mention this aspect in the manuscript.

In the methods section, under Mice and therapies we added: “Using both OCT and slit-lamp examination, no corneal ulcers, surface defects or inflammation was observed in any of the groups of mice.

  1. The authors suggest an increase in ZO-1 levels may be a compensatory response to tight junction disruption (line 348). To clarify and support this statement, it would be beneficial to cite references that align with this theory. Currently, without such references, the interpretation of these contrary results remains unclear to the reader.

We added a reference that observed increased expression of tight junction proteins upon compromise of intestinal barrier function.

  1. In line 355, the authors mention an unknown kinase or transcription factor. For the benefit of the readers and to add depth to your research, please specify or provide more context regarding the unknown kinase or transcription factor. This could include its role in the mechanism you're studying, previous research relating to it, or why its identity remains unknown.

This paragraph is edited with mention of a specific kinase and transcription factor affected by oxidative stress.

  1. Their discussion section appears quite brief and somewhat speculative without substantial evidence to back up their points. To enhance the clarity and depth of the paper, I suggest they move portions of the discussion found in the results section to the actual discussion section. By doing so, they can elaborate on their results more comprehensively, provide more context and interpretation, and tie their findings more explicitly to existing literature. This would also help to streamline the results section, making it more straightforward for readers to follow.

The Results have been streamlined as suggested and brought to the Discussion to further support the strong evidence backing up the conclusions.

Reviewer 2 Report

The manuscript by Ogando, Kim, Li, and Bonanno titled “Corneal edema in inducible Slc4a11 knockout is initiated by mitochondrial Superoxide Induced Src Kinase activation” uses a mouse model to provide new insight into the mechanism behind the corneal swelling induced by lack of Slc4a11, a protein mutated in human congenital corneal dystrophy. The study demonstrates the expertise of the Bonanno lab in the study of this protein. 

A concern with the study is the way data are presented and interpreted.  The data presented in Figure 2 are mentioned (lines 201-203) before all the data in Figure 1 have been presented.  In addition, the data in Figure 4 are described (lines 244-245) in the sentence that precedes the sentence that cites Figure 4.

The authors present data showing the localization of ZO-1 and F-actin in several figures (3, 6, and 7).  The results section describing the data in Figure 3A states that “The integrity of tight junctions (studied by ZO-1 staining) and cortical cytoskeletal (studied by F-actin) were altered at 14 days but not 7 days”.  These claims are not supported by any sort of quantitative analysis in Figure 3 or in Figures 6 or 7.  They should be.  It is not sufficient to show that the levels of ZO-1 are decreased in protein extracts from corneal endothelial cells to justify making a statement about differences seen in ZO-1 staining.

The authors present some data from control as well as day 7 and day 14 after they initiate tamoxifen (Tm) treatment in the animals food; they also present some data only from day 14 comparing a PBS treated to Tm treated corneal tissues. Corneal thickness and fluorescence permeability are not altered at day 7 but are altered at day 14; lactate and MitoROS are elevated at day 7 and day 14 compared to controls.  We are told that lactate transport requires an active osmotic barrier and intact tight junctions.  So, why are lactate and MitoROS elevated in the KO stromas at day 7 if the tight junctions are intact at day 7?  NaK ATPase activity and Src kinase activity are elevated at day 14; we don’t know if these proteins contribute to the elevated lactate and MitoROS seen at day 7. 

Finally, while the data on the impact of Visomitin and eCF506 eyedrops on corneal thickness and permeability presented in Figure 7 are impressive, when compared to controls, the treated eyes do not appear to be entirely normal.  Although there was no comparison made to controls, the fluorescence permeability and stromal lactate levels appear to be elevated compared to controls.  The data are presented for day 14 only. It is important to know what is going on at day 7 when lactate and MitoROS are elevated in the KO corneas.  The data showing ZO-1 localization and F-actin are not quantified; making statement about changes in the localization of these proteins without supporting these claims with quantification is not justified. 

Author Response

Comments and Suggestions for Authors

#2

The manuscript by Ogando, Kim, Li, and Bonanno titled “Corneal edema in inducible Slc4a11 knockout is initiated by mitochondrial Superoxide Induced Src Kinase activation” uses a mouse model to provide new insight into the mechanism behind the corneal swelling induced by lack of Slc4a11, a protein mutated in human congenital corneal dystrophy. The study demonstrates the expertise of the Bonanno lab in the study of this protein.  

A concern with the study is the way data are presented and interpreted.  The data presented in Figure 2 are mentioned (lines 201-203) before all the data in Figure 1 have been presented.  In addition, the data in Figure 4 are described (lines 244-245) in the sentence that precedes the sentence that cites Figure 4.

Thanks for pointing this out. This has been reorganized and clarified to stay in sync with the figures.

The authors present data showing the localization of ZO-1 and F-actin in several figures (3, 6, and 7).  The results section describing the data in Figure 3A states that “The integrity of tight junctions (studied by ZO-1 staining) and cortical cytoskeletal (studied by F-actin) were altered at 14 days but not 7 days”.  These claims are not supported by any sort of quantitative analysis in Figure 3 or in Figures 6 or 7.  They should be.  It is not sufficient to show that the levels of ZO-1 are decreased in protein extracts from corneal endothelial cells to justify making a statement about differences seen in ZO-1 staining.

The quantitative analysis, fluorescein permeability, is shown in figure 3B, 6B and 7B.

The levels of ZO-1 actually increased, the morphology was altered, and the fluorescein permeability increased. The increase in ZO-1 was unexpected, but could be a compensatory response.

The authors present some data from control as well as day 7 and day 14 after they initiate tamoxifen (Tm) treatment in the animals food; they also present some data only from day 14 comparing a PBS treated to Tm treated corneal tissues. Corneal thickness and fluorescence permeability are not altered at day 7 but are altered at day 14; lactate and MitoROS are elevated at day 7 and day 14 compared to controls.  We are told that lactate transport requires an active osmotic barrier and intact tight junctions.  So, why are lactate and MitoROS elevated in the KO stromas at day 7 if the tight junctions are intact at day 7?  NaK ATPase activity and Src kinase activity are elevated at day 14; we don’t know if these proteins contribute to the elevated lactate and MitoROS seen at day 7.  

Lactate can accumulate independently of the changes in the tight junctions, it can be induced for example by decreased expression or activity of the NaK ATPase or MCTs. Our results indicate that the changes in transporter activity are observed before the changes in barrier function.

Finally, while the data on the impact of Visomitin and eCF506 eyedrops on corneal thickness and permeability presented in Figure 7 are impressive, when compared to controls, the treated eyes do not appear to be entirely normal.  Although there was no comparison made to controls, the fluorescence permeability and stromal lactate levels appear to be elevated compared to controls.  The data are presented for day 14 only. It is important to know what is going on at day 7 when lactate and MitoROS are elevated in the KO corneas.  The data showing ZO-1 localization and F-actin are not quantified; making statement about changes in the localization of these proteins without supporting these claims with quantification is not justified.

Because there is no increase in corneal thickness at 7 days, (see Figure 1B), we did not test that time point.

Reviewer 3 Report

Summary:

This is a comprehensive and well-written study describing the contribution by Slc4a11 NH3-activated expression in preserving corneal endothelial homeostasis and tissue transparency in mice. Its role is expertly documented by evaluating the impact of its loss of function on Na-K ATPase-mediated ion pump activity and barrier function.   Tamoxifen feeding for 14 days excised the Slc4a11 NH3 gene, which in turn increased both corneal thickness and tight junctional fluorescein permeability. Previous studies underlie the assumption that the loss of mitochondrial inner membrane Slc4a11 NH3-activated mediated H+ influx led to mitochondrial membrane potential hyperpolarization. This presumed change was associated with rises in mitochondrial superoxide expression levels. Such increases resulted in disruption of junction ZO-1 distribution and cortical cytoskeleton reorganization at day 14 concomitant with decreased expression of claudin1 that resulted from an increase in its tyrosine phosphorylation. Stromal lactate accumulation increased by 60%, and Na+-K+ATPase activity decreased by 40% after 14 days. Such effects were accompanied by significant declines in the expression of lactate transporters MCT2 and MCT4. Rises in oxidative stress-induced increases in Src kinase that compromised pump function and barrier intactness. Treatment with a mitochondrial antioxidant, Visomitin (SkQ1, mitochondrial-targeted antioxidant) or Src kinase inhibitor eCF506 significantly slowed the increases in CT, and decreased concomitant rises in stromal lactate retention, improved barrier function, reduced Src activation and claudin1 phosphorylation, and rescued declines in MCT2 and MCT4 expression. There are several clarifications that would measurably improve the impact of this relevant study.

Concerns:

1.     The results of the Slc4a11 NH3 knockout model confirm that oxidative stress is an important factor in the pathogenesis of Fuchs endothelial corneal dystrophy (FECD). It would be informative to compare its phenotype and the underlying pathophysiological events with those described in another FECD menadione model since the initial change in the inner mitochondrial membrane voltage accompanying rises in ROS generation was not the same. In the current study, the membrane voltage is assumed to be a hyperpolarization based on a previous study from the same group whereas in another study using a menadione model the measured membrane voltage underwent depolarization.  The earlier study from the same group is PMID 35053313 and the menadione study is PMID 26935406. It will be informative to confirm through measurements of the membrane voltage that the membrane voltage underwent hyperpolarization in the NH3 knockout model.  

 2.     The phenotype of the SLC 4A 11 knockout was previously described and found to be less severe than in the current study PMID 19586905. Please clarify whether there is any basis for this difference.

 3.     Need to cite the original publication showing that oxidative stress is an inciting factor of the pathophysiological processes of FECD. PMID 20847286 and 11850437

Author Response

Comments and Suggestions for Authors

#3

Summary:

This is a comprehensive and well-written study describing the contribution by Slc4a11 NH3-activated expression in preserving corneal endothelial homeostasis and tissue transparency in mice. Its role is expertly documented by evaluating the impact of its loss of function on Na-K ATPase-mediated ion pump activity and barrier function.   Tamoxifen feeding for 14 days excised the Slc4a11 NH3 gene, which in turn increased both corneal thickness and tight junctional fluorescein permeability. Previous studies underlie the assumption that the loss of mitochondrial inner membrane Slc4a11 NH3-activated mediated H+ influx led to mitochondrial membrane potential hyperpolarization. This presumed change was associated with rises in mitochondrial superoxide expression levels. Such increases resulted in disruption of junction ZO-1 distribution and cortical cytoskeleton reorganization at day 14 concomitant with decreased expression of claudin1 that resulted from an increase in its tyrosine phosphorylation. Stromal lactate accumulation increased by 60%, and Na+-K+ATPase activity decreased by 40% after 14 days. Such effects were accompanied by significant declines in the expression of lactate transporters MCT2 and MCT4. Rises in oxidative stress-induced increases in Src kinase that compromised pump function and barrier intactness. Treatment with a mitochondrial antioxidant, Visomitin (SkQ1, mitochondrial-targeted antioxidant) or Src kinase inhibitor eCF506 significantly slowed the increases in CT, and decreased concomitant rises in stromal lactate retention, improved barrier function, reduced Src activation and claudin1 phosphorylation, and rescued declines in MCT2 and MCT4 expression. There are several clarifications that would measurably improve the impact of this relevant study. 

Concerns:

  1. The results of the Slc4a11 NH3 knockout model confirm that oxidative stress is an important factor in the pathogenesis of Fuchs endothelial corneal dystrophy (FECD). It would be informative to compare its phenotype and the underlying pathophysiological events with those described in another FECD menadione model since the initial change in the inner mitochondrial membrane voltage accompanying rises in ROS generation was not the same. In the current study, the membrane voltage is assumed to be a hyperpolarization based on a previous study from the same group whereas in another study using a menadione model the measured membrane voltage underwent depolarization.  The earlier study from the same group is PMID 35053313 and the menadione study is PMID 26935406. It will be informative to confirm through measurements of the membrane voltage that the membrane voltage underwent hyperpolarization in the NH3 knockout model.  

Our 2019 Redox Biol paper showed mitochondrial hyperpolarization of Slc4a11 KO endothelial cells that preceded significant depolarization. The absence of the glutamine sensitive Slc4a11 uncoupler leads to hyperpolarization, which then increases mitoROS, damages the mitochondria, leading to mPTP opening and cell apoptosis.

  1. The phenotype of the SLC 4A 11 knockout was previously described and found to be less severe than in the current study PMID 19586905. Please clarify whether there is any basis for this difference. 

The publication that the reviewer is referring is Lopez et al, J Biol Chem 2009 (PMID 19586905) that describes a Slc4a11 KO mouse (conventional= not inducible) with very little effect on the corneal endothelium.  Later, Han et al, IOVS 2013 created a different Slc4a11 KO model (conventional) that causes significant corneal edema (PMID 23942972). The two groups used different KO approaches, yet it is not clear why the phenotypes models differ. We obtained the Slc4a11 KO from Dr Vithana (Han was a member of the Vithana group) and used this model in previous publications (PMID: 31254733 and PMID: 34871568). The present paper uses an inducible Slc4a11 KO that we established recently (PMID: 34190974) and has a similar phenotype to the conventional KO.

  1. Need to cite the original publication showing that oxidative stress is an inciting factor of the pathophysiological processes of FECD. PMID 20847286 and 11850437

We agree. We added in discussion a sentence “Oxidative stress is main contributor in the pathophysiological process of FECD and CHED” and included the mentioned references.

Round 2

Reviewer 1 Report

The revised manuscript provided by Ogando and co-authors has effectively addressed the comments from the previous review.

Reviewer 2 Report

The authors acknowedge this reviewers concerns.